# Environmental Management of Equine Asthma

**DOI:** 10.3390/ani14030446

**Published:** 2024-01-30

**Authors:** Elisa Diez de Castro, Jose Maria Fernandez-Molina

**Affiliations:** 1Veterinary Teaching Hospital, University of Cordoba, 14014 Córdoba, Spain; 2Department of Animal Medicine and Surgery, University of Cordoba, 14014 Córdoba, Spain; 3Department of Physical Chemistry and Applied Thermodynamics, University of Cordoba, 14014 Córdoba, Spain

**Keywords:** horse, equine asthma, stabling, environment, therapy, dust, soaking, bedding

## Abstract

**Simple Summary:**

Equine asthma, a chronic respiratory disease in horses, is primarily caused by environmental factors. In this review, we will present and compare the most relevant and recent information about the environmental management of equine asthma. In terms of feeding, not only the type of forage but also its production method and potential contamination are the most important factors to consider. Soaking and steaming forage can improve its hygienic quality, but these practices also reduce its nutritional value, necessitating dietary supplementation. Concerning housing, despite some contradictory results, avoiding straw bedding and improving barn ventilation remain common recommendations when pasture turnout is not possible. Finally, owners’ compliance has been identified as the most crucial aspect of effective environmental management and the focus of equine asthma treatment should be a detailed explanation to owners of the real benefits of these measures.

**Abstract:**

Environmental practices related to the inhalation of airborne dust have been identified as the main cause of equine asthma (EA) and reasonably, they are truly relevant in its treatment and control, especially for horses with its severe form. Vast research regarding environmental recommendations has been conducted in recent years. However, no recent exhaustive reviews exist that gather all this new evidence. The aim of this review is to report and compare the most pertinent information concerning the environmental management of EA. The main findings highlight the importance of the type of forage used for feeding but also its method of production and possible contamination during manufacture and/or storage. Procedures to reduce this, such as soaking and steaming hay, improve its hygienic quality, although they also decrease forage’s nutritional value, making dietetic supplementation necessary. Regarding stabling, despite some conflicting results, avoiding straw as bedding and improving barn ventilation continue to be the common recommendations if turning to pasture is not feasible. Finally, owners’ compliance has been identified as the most critical point in correct environmental control. Educating owners about the genuine benefits of these measures should be a cornerstone of EA management.

## 1. Introduction

Equine asthma (EA) is a chronic non-infectious inflammatory disease of the lower airways that is characterised by respiratory signs such as a cough and increased respiratory effort. The nomenclature of this disease has recently changed. Two conditions previously considered to be different clinical entities, recurrent airway obstruction (RAO) and inflammatory airway disease (IAD), are now denominated as EA (severe or mild, respectively). They have similar clinical presentations, but many differences may exist in causes, severity, and pathologic characteristics [1,2,3]. Furthermore, clinical signs may vary seasonally and worsen with exposure to trigger factors [4,5,6,7].

There is a large body of evidence that supports environmental and feeding practices as the main cause of asthma [4,8,9,10,11,12]. Some of the most important factors considered include air quality, ventilation, hay, mould, endotoxin, and bedding [13]. The impact of air quality on equine asthma is determined by the number, size, and types of particles present in the air, with ventilation quality playing a crucial role. Particles measuring less than 5 μm in diameter, including endotoxins, moulds, and microbial toxins, are potential inflammatory agents found in stable dust. However, the relative significance of each component in the pathogenesis of equine asthma remains unclear. Some of these particles have been shown experimentally to induce a neutrophilic inflammatory response and associated pulmonary dysfunction. Existing knowledge supports a hypersensitivity component linked to these particles in equine asthma, though the underlying immunological mechanisms involved are particularly complex [14].

Nevertheless, there is another type of EA that, instead of being associated with stabling, appears in horses on pasture (formerly known as pasture-associated obstructive pulmonary disease). In this case, the trigger factors for airway inflammation are suggested to be pollens and fungal spores [10]. Environmental recommendations are probably not applicable to this variant of the disease, and hence, its management will not be addressed in this review.

Several investigations have focused on the possible relationship between viral infections and the development of asthma. Some evidence suggested the presence of some virus such as equine herpesvirus 2 and 5 or equine rhinovirus in horses with mild asthma [15,16]. However, no evidence of causality has been found between them.

Diagnosis is commonly made by tracheobronchial endoscopy and cytology of respiratory secretions, but other techniques like the study of respiratory function, blood gases analysis or the measurement of different biomarkers can be helpful in characterising the disease and its severity.

Treatment of horses with asthma aims to decrease bronchospasms and associated coughs and to reduce lower airway inflammation and mucus production, especially in the prevention of new episodes. Two main therapeutic approaches have traditionally been considered: environmental management and pharmacological treatment. The latter focuses on controlling airway inflammation through corticosteroid therapy and reducing bronchospasms with bronchodilators. However, it is now widely established that appropriate environmental management is the most important approach to treat airway inflammation and dysfunction in horses with asthma, both in severe and mild cases [17,18,19]. Thus, the modification of horses’ environment has become the centre of most prevention and treatment strategies. Changes in the type of bedding and forage in stabled horses are the mainstay of those approaches, and maintaining horses at pasture to reduce exposure to airborne triggers is preferred [20,21,22].

To our knowledge, no exhaustive reviews about the environmental management of EA exist in recent literature. Due to this fact, information should be extracted from different types of articles that often have other primary objectives, such as validating diagnostic methods. We believe that it is essential for clinicians to have access to a comprehensive overview of this information, as this would simplify the process of explaining these strategies to horse owners. In this review, we will assess and evaluate the most relevant information regarding the treatment and prevention of EA with environmental changes.

## 2. Review Methodology

Searches were conducted using the electronic databases NIH Pubmed, Scopus, and Web of Science (WOS) during the month of September 2023. The chain used in search engines was “(horse* OR equine OR Equidae or Equus) AND (asthma* OR “recurrent airway obstruction” OR heaves OR COPD OR “chronic obstructive pulmonary disease*” OR “inflammatory airway disease” OR IAD OR RAO OR “broken wind” OR “small airway disease*”) AND (“antigen avoidance” OR “environment*” OR prevent* OR “low AND dust” OR “hay”)” and only scientific articles written in English, Spanish, and French were eligible.

The first search included a total of 1280 references: 430 in Pubmed, 415 in WOS, and 426 in Scopus. The time range was restricted to the last 20 years (2003–2023) and categories not related to veterinary medicine and agricultural and biological sciences were eliminated. After the removal of duplicates using the reference manager Mendeley, 111 articles remained. Firstly, refinement was implemented after examining research titles and excluding articles of other species and other aspects of EA such as pathogenesis, diagnosis, or pharmacologic treatment only. Secondly, after an exhaustive reading of all the abstracts, 54 final articles were selected. Finally, 22 additional references were obtained through the snowballing technique to support the information presented, obtaining a final number of 76 references.

There is a high heterogenicity among the studies included in this review, which affects several aspects of them. For example, healthy horses, horses with natural disease (mild or severe asthma) with or without matched controls, and horses with induced disease are all included. Likewise, diverse approaches to evaluate the influence of environment on respiratory function are used, such as clinical score, arterial blood gases, airway cytology, bronchial endoscopy, different pulmonary function tests, or even competition results, to evaluate performance. In addition, methods for assessing the quality of air, e.g., particulate matter, presence and quantification of endotoxin, moulds, volatile gases, etc., differ among studies. This variability complicates the comparison of results among studies and therefore the extraction of conclusions. Consequently, a narrative review has been chosen as the best approach to assemble the current information about the topic of this publication.

## 3. Importance of Environment

Equine asthma is caused or exacerbated by the inhalation of airborne dust. Organic dust is very abundant in stables. It contains a great variety of substances rich in potential sources of allergens such as moulds, endotoxins, ß-glucans, or mite and plant debris, among others, which are all capable of inducing airway inflammation [3,7,8,23,24,25] In fact, the sole fact of stabling an asthmatic horse can cause an acute exacerbation of the disease and an increase in cortisol after only six hours [26].

Horses with asthma respond to environmental challenges (stabling and hay feeding) by sterile inflammation with alterations in bronchoalveolar lavage cytology [11,27,28]. Changes mainly include neutrophilic inflammation and mucus accumulation in the lungs. Nevertheless, they are not exclusive to asthmatic horses, as healthy horses exposed to the same trigger factors also develop inflammation [3,27,28,29].

Several indices have been developed to evaluate airborne dust, such as the respirable dust concentration (RDC): the portion of dust small enough (<5 µm) to enter the small airways [25]. Particle size may determine the impact of dust on airway inflammation as larger particles, known as “inhalable” (<10 µm or PM10), generally do not reach the lower airways. Nonetheless, small particles, or “respirable” (<5 µm or PM5), such as mould spores, will reach lower airways and thus could cause airway inflammation [23,25,30,31]. Although this classification is commonly used in most studies, the exact size of the particles that can reach equine lower airways is not known. Despite this, PM5 has been associated with eosinophilic inflammation in young thoroughbreds [32] and with neutrophilic inflammation in actively racing thoroughbreds [3,31] so it can support the use of PM categorisation. However, one of the concerns related to the importance of these small respirable particles is that they are more difficult to identify by the owners, as they perceive their stables to be “clean”. A clear explanation of this fact will help to improve their compliance with treatment.

Several hay moulds such as *Aspergillus fumigatus, Faenia rectivirgula, and Thermoactinomyces vulgaris* have been implicated in EA pathogenesis [6,33,34]. Endotoxins, on the other hand, apart from being significantly higher in stables compared with pasture [35], increase the inflammation caused by other agents but they alone do not induce severe clinical signs [36].

Two observational studies investigate the impact of environmental factors (dust, endotoxin, and ß-glucan exposures) in different populations [3,12]. The first one evaluated a group of 64 thoroughbred racehorses in 98 examinations to detect the presence of mild EA and its relation with their performance [3]. Most of those horses (80%) were diagnosed with mild asthma by bronchoalveolar fluid (BALF) cytology performed one hour after racing, although they were theoretically healthy horses. Additionally, an increase in mast cells and neutrophils was related not only to RDC (neutrophils) and ß-glucan exposure (mast cells) but also to a decrease in performance, which supports the necessity of implementing strategies to reduce dust exposure. This study also sustains the previous idea that stall confinement is enough to induce airway inflammation. 

In the second study, the influence of air quality on respiratory clinical signs or biomarkers of airway inflammation was assessed in 12 standardbred trotters. Their results show that, even when horses were bedded on straw, levels of respirable dust and organic dust were lower than recommended limits for humans and horses. One reason for this inconsistency may be that samples, different to other studies, were taken from outside the boxes [12]. This is one of the problems of comparing studies that assess total and respirable dust because the results of measurements depend enormously on the zone from which they are taken. For instance, dust in the breathing zone of a horse is always higher than in samples obtained from a fixed point in the barn [37]. For that reason, a more standardised method of measurement should be employed to properly compare the results of different studies.

There are various methods to quantify dust in stables. Clements and Pirie [25] compared the traditional and most popular technique for measuring airborne dust concentration (ADC) with a real-time continuous particle monitor. The first method involves collecting dust onto pre-weighed filters and then calculating the mean ADC. One of its advantages is the ability to obtain dust samples for future analysis. The second one, apart from calculating the mean ADC, allows the identification of punctual elevations in it, which could be harmful to horses with respiratory inflammation. That comparison indicates a good agreement between both procedures. 

Several factors determine airborne microorganisms’ concentrations inside the stables. Feed apparently has a more significant impact on airborne content than bedding [25], as changing from hay to haylage reduces the mean RDC by about 70%, compared to a 30% reduction with a change from straw to wood shavings [23]. One possible reason for this is the closer proximity of the nostrils to forage compared with bedding. However, another study [37] reports that using straw as bedding material further increases the ADC, compared to hay. Those discrepancies could be due to multiple factors, mainly to the different procedures used for measuring dust (RDC vs. ADC), but also to the quality of both straw and hay, the method of storage, or different barn architecture.

In summary, airborne dust is abundant in stables and inherent to the actual management of stabled horses. Respirable particles, such as mould spores, cause airway inflammation in horses. However, the recommended limits of these particles for horses should be taken cautiously, as the method and particularly the localisation of the measurement selected will influence its results. Although the factors of feed and the stable’s bedding and management generally have an additive effect over airborne dust, with the purpose of this review, they will be addressed separately.

### 3.1. Feeding

Evidence indicates that feeding, and forage in particular, exerts a greater influence on respirable dust levels than the type of bedding [25,38]. Hay is the most common source of fibre in stabled horses, and it is also used to complement the feeding of horses in pasture. Notwithstanding, hay consumption has been associated with airway inflammation both in healthy and asthmatic horses [5,13]. This may be due to its high concentration of dust particles, moulds, fungal spores, endotoxins, or other compounds related to the technique chosen to produce hay [39,40].

The production of hay requires dry weather to reduce moisture content to 15–20%, which requires time and adequate weather conditions. If hay is not dry enough before baling it, mould spores that contaminate the hay during harvest will proliferate. On the other hand, alternative forage choice such as haylage only requires decreasing moisture to about 50–70% before being wrapped in anaerobic conditions to preserve it [41], and that could be performed faster. High care must be taken nevertheless not to puncture plastic bags to avoid moisture and/or air entering them, which could support fungal growth. The method used to store forage and straw can also influence fungal proliferation [40]. Another concern about haylage is the potential risk of contamination with *Clostridium botulinum* if it is not properly preserved. To decrease this risk, bales should be examined and discarded if signs of mould growth are suspected. Additionally, haylage should be fed within 3–7 days of opening [42], and this can cause practical issues for the owner if not many horses are receiving this kind of feed.

#### 3.1.1. Forage Quality Analysis

Despite the fact that dusty hay has been involved in the pathophysiology of EA, information about its composition is often not addressed in EA studies. The initial quality of hay, storage, methods of feeding, and any treatment applied to it affect its hygienic quality and thus the number of dust particles a horse may inhale [34,43].

The hygienic quality of hay can be improved, for example, by soaking and/or steaming it. Various protocols have been described for both methods [44,45,46] but a consistent recommendation for their application remains critical. In essence, soaking involves a full immersion of hay in water for a predetermined duration (15–30 min) prior to feeding it to the horse. For steaming, a steam generator directly delivers steam into a closed compartment where the hay is subjected to high temperatures, approximately 100 °C, for a minimum of one hour to prevent detrimental consequences like the proliferation of new bacteria. These processes lower the number of small particles (soaking) and mitigate microbial contamination (steaming) in the hay [43,44,45].

With regard to this matter, several authors have compared the effects of diverse managements on the respirable particle and microbial content of hay (Table 1). Moore-Colyer et al. [46], compared ten minutes of soaking and three different devices for steaming with this purpose. They concluded that a high-temperature steamer is not only the best to decrease particles (99% vs. 88% in partial steaming), but it also nearly eliminates microbial contamination while other systems did not reduce it. For that reason, those cannot be recommended to improve the hygienic quality of hay. In a study by Glatter et al [40], soaking and steaming hay are compared, as well as the effect of the hay’s subsequent storage on hygienic quality. Their results showed that both methods achieve a reduction in microorganisms and moulds. However, the higher moisture of forage after soaking makes its storage unstable and prone to the proliferation of microorganisms. For that reason, feeding soaked hay should only be considered a suitable option if it is used immediately.


Several studies have investigated the presence of fungi in horses’ environments [11,39,43,49]. For example, one analysed the presence of fungi (*Aspergilus and Fusarium spp.*) and mycotoxin contamination in various types of feed, three forages and two concentrates [49]. *Aspergillus* was the most common isolate both in hay and haylage, although considerable differences in contamination existed between forage harvested in Ireland, where the weather had been excessively rainy, and that produced in Canada (50% contamination vs. 13%). On the other hand, the levels of fungi in concentrate were significantly lower than in forages, and pelleted feed contained the lowest number of pathogenic fungi. Regarding mycotoxins, ochratoxin and fumonisin were the most frequently found in both forages and concentrates. Other studies that focus on the influence of feeding concentrate on airway particulate matter conclude that this effect depends on the type of concentrate, technical treatments of them [50] and supplementation with additives like oil or molasses [51,52].

Feeding dry hay, as shown in a recent prospective observational study of 731 horses, increases the probability of having fungi in a tracheal wash and the diagnosis of mild asthma in sport horses [11], and hence, this product should not be recommended for them. Likewise, significantly higher levels of mould have been found in hay compared with haylage and this difference persisted after soaking, although mould counts in hay decreased [43]. On the other hand, in a study by Dauvillier et al., eating steamed hay decreased fungi content in equine airways, while soaked hay, haylage, or “dust free” hay did not have the same effect [11].

A hygienic analysis was performed by Intemann et al. [41] comparing microbiological counts and endotoxin levels among hay, straw, and haylage samples. The results of the study did not find significant differences in contamination with moulds among forages, although significantly more yeast and bacteria were frequently found in haylage. However, the dry matter content of haylage samples was inadequate in more than 60% of the cases, which can also influence mould counts. On the other side, high levels of endotoxins are most frequently found in straw, as reported previously [38]. When the reason for analysing the forage was coughing after feeding, the results showed a clear relation with the presence of *Aspergillus spp.* but not with other dust compounds. As *Aspergillus* has been more likely to be detected in hay samples, this kind of forage may produce greater health risks to horses than other forages.

Although the possible harmful effect of fungi in asthma is not completely understood, as it can be allergenic, infective, toxic, or any combination of them, the association between *Aspergillus spp.* and coughing may alert us to the potentially harmful effects of at least some of them. Therefore, we consider that the role of fungi in the pathogenesis of IAD and equine asthma should be further explored.

As a result of those investigations, we can assume that forages that are not well prepared or well conserved are at a higher risk of becoming contaminated with moulds or microbes. Contamination of forage is relatively abundant, which puts horses at risk of developing respiratory inflammation unless hygienic measures are taken.

A conclusion that can be extracted from all the research conducted in this field is that the key point of forage quality is how it is produced, as it enormously influences the characteristics of forages. Nevertheless, it is often beyond the direct control of individual horse owners. Recommendations when forage quality is doubtful include the employment of steamed hay (with a high-temperature standard steamer), hay soaked immediately before feeding, haylage, or pelleted food. However, there is still a lack of information about the ideal protocol needed to improve forage quality without undesirable consequences. Additional studies comparing the standard protocols of soaking and steaming hay specifically regarding RDC and fungi are necessary.

#### 3.1.2. Effects of Hay Treatments over Its Nutritional Quality and Palatability

The benefits of soaking or steaming hay on its hygienic quality have been extensively described. However, less information is known about their impact on forage nutritional quality and acceptance by horses (Table 2).

Martinson et al. [53] evaluated the influence of water temperature and time of soaking on nutrient loss from two kinds of forages, alfalfa and orchard grass hay, as well as different cuts of them. The conclusions of the study indicate that nutrient losses vary depending on different soaking times, types, and cuts of hay, which enormously complicates their interpretation. Nevertheless, based on their results, 15–60 min of soaking would not cause nutrient deficits for horses in light work, although supplementation of crude protein (CP), Ca, P, K, and Mg could be necessary for horses with higher necessities, such as heavy exercise or early lactation. A more recent study by Bochnia et al. [54], investigates the effects of different lengths of soaking, not on crude protein as in previous studies but on pre-caecal digestible crude proteins (pcd CP) and amino acids (pcd AA). The authors conclude that soaking hay for 15 min is sufficient to reduce the levels of several important nutrients such as fructans, water-soluble carbohydrates, macronutrients, and trace elements (P, K, Mg, Zn, Mn, Cu, and Fe). Additionally, pcd CP and pcd AA also significantly decreased and metabolic energy was also reduced by 5–15%.
animals-14-00446-t002_Table 2Table 2Effects of hay treatments (compared with dry hay) on its nutritional quality and acceptance.
Type of Treatment
Soaked HaySteamed HayMineral contentReduction in phosphorus, potassium, sodium, magnesium, zinc, manganese, copper, and iron [54]. Slight decrease in calcium [40] Minor reduction in phosphorus levels [41,48]. Slight decrease in Ca [40]ProteinsReduction in pre-caecal digestible crude protein and amino acids. [54]. No clear reduction in same parameters in another study [40]Considerable reduction in pre-caecal digestible crude protein [40,54]. Increase in insoluble part of crude protein [54,55]FibreIncrease in crude fibre content [40]No significant changes in fibre content [48]AcceptanceReduced palatability and restrained feed intake reported [40]Generally well accepted, with increased consumption compared to dry or soaked hay [40]


Earing et al. [48] evaluate, in a crossover design, the effects of hay steaming on grass and alfalfa hay with different degrees of mould content. Steaming hay reduced its dry matter (DM) by increasing moisture but also reduced P content. On the contrary, crude protein, acid (ADF), neutral detergent fibre (NDF), and starch content of the hay were not affected by steaming. However, temperatures observed during the steaming process in their study did not exceed 50°C and therefore, changes in fibre fraction concentrations were not expected. A study by Moore-Colyer et al. [46] also showed that after steaming 30 different samples of hay, there was no reduction in CP, Ca, Mg, Na, P, Cu, Mn, N, K, and Zn compared with non-steamed hay. To evaluate more deeply the potential effects of steaming on protein damage, Pisch et al. [55] analysed six batches of hay and compared them before and after being steamed at high temperatures. Their results showed that steaming did not affect CP content as previously shown [47,48]. However, it increased the insoluble part of CP and decreased the pcd CP and some essential amino acids, similar to what occurred with soaking in a previously mentioned study [54]. Despite this similarity, the mechanism of this wastage is different to the case of soaking, as in steaming seems to be related to heat damage [55], while the decrease after soaking is caused by nutrient wash-out [54].

Likewise, Glatter et al. [40], compared the effect of several hay treatments (soaking and steaming) on nutrient content. Crude fibre increased after soaking and decreased after steaming hay, and water-soluble carbohydrates decreased after soaking but increased after steaming. Concerning minerals, Ca levels slightly decreased with both treatments and P, K, and Na decreased in soaked hay. A considerable reduction in soluble protein, non-structural carbohydrates, and K was also found after soaking hay in a recent study, which also presented some differences in nutrient losses. These discrepancies are probably due to the selection of a different initial forage [56] The authors considered that this loss could be significant in racehorses due to their high nutrient demands. On the other hand, steamed hay is considered a good option to conserve nutrients while reducing airborne dust.

Several methods exist to address the acceptance of the feed by the horse, for instance, studying the feed intake [44,56] or the chewing activity of the horse [40]. In the latter study, all horses accepted both soaked and steamed hay although individual preferences existed. Soaked hay, however, seemed to have a reduced palatability because horses showed a restrained feed intake. Earing et al. [48] also evaluated dry matter intake and concluded that horses consumed more steamed hay than dry hay, but differences existed only if the original hay had a low quantity of mould. Similar results were reported by Owens et al. [56] in a methodical study that compared feed preferences among soaked, steamed, and dry hay, and they found that horses consumed less soaked timothy-alfalfa hay than dry or steamed hay.

Clinicians and owners should be aware that soaking, and to a lesser extent steaming, can alter the nutritional value of hay, although great variability exists among different types of hay. Overall, the described studies highlight the importance of considering the impact of different treatments on the nutrient content of hay for horses, particularly mineral and protein losses. However, the diverse methodologies employed and the initial forage used can lead to variable results and conclusions. For instance, temperature ranges and soaking durations differed between studies, influencing the extent of nutrient depletion. There is a consensus about the need for dietary supplementation when steamed or especially soaked hay is fed. Furthermore, it cannot be ignored that receiving a diet with protein deficits could negatively affect equine growth and performance. Therefore, it is our responsibility as veterinarians to alert owners of this possible complication of treating hay when no diet modification is implemented. Improvement in knowledge regarding diet adaptations for horses eating soaked or steamed hay is therefore fundamental. 

The referenced articles provide valuable information on feed acceptance in horses, describing that soaking hay reduces its palatability; thus, it is less consumed than dry or steamed hay. These results, considered together with the previous ones, should alert us that feeding only soaked hay is probably insufficient for horses with high nutritional needs. Future research in this area would benefit from standardised methods of assessment, ensuring the comparability of results across studies. Additionally, the inclusion of control groups and blinding, which is lacking in some of them, would improve the strength of the findings.

### 3.2. Bedding and Environmental Management

Proper bedding is essential when a horse lives indoors. As mentioned previously, it has a significant effect on air quality, although some controversial information exists about the influence of different types of bedding on airway inflammation (Table 3).

Information obtained from various studies [25,57] states that there is a higher concentration of dust when straw is utilised as bedding compared with wood shavings or other materials: paper, cardboard, or peat moss. Older studies comment that the conservation of materials also has a high impact on their hygienic quality, and with proper storage, differences between straw, wood shavings, and paper bedding decrease [58]. Wood pellets and straw pellets are bedding types with lower particle counts, and according to Fleming et al. [57], they are thus especially suitable for horses suffering from chronic and allergic ailments of the airways.

Comparison among straw, peat with shavings, and crushed wood pellets in a recent study [59] resulted in a lower dust content in peat and shavings compared with crushed wood pellets, although bacterial contamination was lower in the latter. Fungal contamination was also lower in wood pellets than in straw and peat with shavings. Changes in bedding (straw to wood shavings) but also in feed (hay to haylage, or soaking/steaming hay) have been evaluated altogether. These modifications cause a significant reduction in mean and maximum RDC and endotoxin levels [23,25,37,38,60], not only in the box of the changes, but also in the neighbouring stable [60]. Ionisation, a treatment that is believed to reduce dust and microbes by the agglomeration of smaller particles into larger ones, did not show an effect on dust quantity or air quality [23].

In summary, a general consensus is that traditional straw bedding is not suitable as bedding for asthmatic horses and alternatives should be used. Among them, wood shavings are considered a good choice, although in some cases, their effects have been evaluated together with a change of feeding [37]. Other described suitable options include paper, cardboard, peat, or wood pellets. In general, inorganic beds have a lower number of microorganisms, despite a higher content of dust in some cases. Hence, one important point to further investigate is the relative importance of dust content compared to fungal or bacterial contamination in EA. This will help us to decide the best bedding material option.

Not only inadequate bedding is an important trigger factor of EA [11,25,61] but also other aspects of the environment, such as the type of horse management system, horses` health, number of horses, or feeding schedule, affect air quality [31,62,63].
animals-14-00446-t003_Table 3Table 3Comparison of hygienic characteristics of different bedding materials used in stabled horses.
Type of Bedding
StrawWood ShavingsWood Pellets and Straw PelletsCrushed Wood PelletsPeat with ShavingsPaperDust exposureHigher concentration of dust compared with wood shavings or other materials [25,37,57,59]Reduction in mean and maximum RDC after changing from straw (also feed modification) [23,25,37,38,60] Lower particle counts than straw [57] Higher dust content than crushed wood pellets and straw [59] Lower dust content compared with crushed wood pellets and straw [59] Lower than straw, no significant differences with wood shavings [57]Fungi presenceHigher presence than wood shavings [23], crushed wood pellets and paper [57,59]Significantly lower than straw (also feed modification) [23] Similar counts to straw [57] Lower fungal contamination than straw and peat with shavings [59] Similar fungi presence to straw [59] Lower count of mould fungi in paper compared with straw [57] EndotoxinHigher levels than other forages (also feed modification) [38,60] Significantly lower than straw (also feed modification) [23,38] NRNRNRNRMicrobiological countsHigher bacterial contamination than wood pellets but similar to peat with shavings [59] NRLower counts than straw, like wood shavings and paper [57] Lower bacterial contamination compared with peat with shavings [59] Similar bacterial presence to straw [59] Lowest count of microorganisms in paper compared with straw, and wood shavings [57] NR: not reported in the consulted bibliography.


Recent research [62] investigated air contamination in different kinds of stables in Poland. Their results showed higher contamination in common stables or runners compared with box stables, although different feed was also administered to them. Airborne particles were also compared in stabled horses during distinct seasons (winter vs. summer). Stabling during winter was associated with higher dust exposure, higher neutrophil percentage in BALF, and an upregulation of IL6 mRNA. In summer, conversely, endotoxin levels were higher [12]. Comparison of this or other studies about the effect of seasonality on airborne dust in airway inflammation is nevertheless unrewarding, as climate and management systems in each season vary considerably depending on geographic areas or individual preferences. Nonetheless, high humidity and temperature may deteriorate the clinical status of horses with severe asthma, probably related to their effect on inhalable pollens and moulds [10].

The importance of different activities in stables has also been investigated by several authors, who stated that the maximum RDC measurements were produced during mucking out, grooming, and peak stable activity [9,25,62]. Even inhalable dust in stables with low-dust bedding, e.g., peat or wood shavings, was only above the detection limit during periods of mucking and adding new bedding of both materials [64]. Other minor differences in management can reduce air quality. Among them, feeding horses from a hay net results in more than a fourfold increase in exposure to respirable particulates (PM10) compared to feeding them from the ground [9,17]. However, when hay nets are hung outside the box, results are similar to feeding from the floor [3]. Also, positioning and closure of doors can affect PM10 levels by increasing them up to 30%. Finally, dusty spaces such as indoor arenas may produce very high levels of respirable particulates [50], even 20 times more than what it is described to induce respiratory dysfunction in humans [17]. For that reason, asthmatic horses should only be taken to riding arenas that have been unused for several hours [51].

The general recommendation of removing all horses within a common airspace during mucking out and grooming, together with avoiding feeding asthmatic horses from a hay net, seems reasonable. Nevertheless, it is also important not only to reduce particulate concentration during punctual dusty activities but also throughout the day [31]. The best manner to do so, apart from reducing small particles in food and bedding, is to maintain good ventilation. Despite that, ventilation alone will not suffice if no changes are made in food and bedding, as it is impossible to remove all the harmful particles in a dusty environment. Therefore, a holistic perspective that considers all these elements will be necessary to improve asthmatic horses’ environment.

## 4. Effects of Environmental Changes on Lung Function

When horses live in a low-dust environment, respiratory inflammation decreases, and in the cases of asthmatic horses, it is not necessary to treat them medically for a long time. However, exposure to allergens in these horses, such as stabling again, can cause airway obstruction and inflammation [19]. Although the effects of environmental changes on airborne quality have been extensively studied, information about their effects on the lower airways is not equally abundant in the literature.

### 4.1. Effects of Feeding Changes

Differences between the effects of hay and other kinds of feed on airway inflammation have been investigated [11,41,47]. Steaming and soaking can improve the hygienic quality of the hay [40], by decreasing microbial contamination and the presence of alveolar particles [46]. Nevertheless, it is also important to know if these modifications also have an effect on horses affected with asthma.

Several studies have expanded on the potential benefits of feeding steamed hay. Blumerich et al. [13] compared the effect of feeding steamed or not-steamed hay on the airway response in severely affected asthmatic horses. Horses that ate hay had deteriorated clinical signs and an increase in tracheal mucus compared with the ones eating steamed hay. However, there is no significant effect of steaming on pulmonary function or BALF cytology, although the small number of horses (6) and the short period of study employed may have influenced those results. Later, Orard et al. [65] also evaluated the effect of feeding steamed hay to severely asthmatic horses. They compared six asthmatic horses and six controls and fed them steamed or dry hay for five days. Similarly to the previous study, horses that ate steamed hay showed lower mucus scores but there was no influence of feeding on clinical score and again BALF cytology or cytokine mRNA expression. The authors of those studies concluded that their findings did not support the use of steamed hay as a non-medical therapy for severely asthmatic horses. However, considering the effects of steaming on the quantity of dust, with a reduction in respirable particles of around 100% [46], as well as its effects on clinical signs, and mucus accumulation, we suggest that further investigation is warranted to determine the potential benefits of its use.

Regarding the use of haylage, Olave et al. [47] evaluated, in a pilot study, the influence of feeding hay or haylage for six weeks on dust exposure, airway cytology, and BALF cytokines of horses in training. Their results showed that dust and ß-glucan exposure were lower in the haylage group compared to the hay group. None of the horses developed any signs of respiratory disease, but horses eating haylage had a decreased number of neutrophils until the end of the study. Although it is a small study with only seven horses, it shows that modifications in feeding affect the quality of air. Additionally, a decrease in pulmonary neutrophilia was found after only two weeks of changing [47].

The same authors performed an interesting and very recent study in stabled racehorses to evaluate the differences in dust exposure and airway cytology between different kinds of feed [42]. Specifically, the impact of feeding hay, haylage, or steamed hay on those parameters was compared. They also measured plasmatic levels of Omega-3 fatty acids, a precursor of specialised pro-resolving mediators (SPMs), which could be involved in returning tissue to homeostasis after an inflammatory process. Horses eating haylage had lower neutrophils than baseline and horses eating hay. Mast cells were also lower at three weeks but returned to baseline at week six. Steamed hay, however, did not decrease BALF neutrophils or cause changes in mast cells. Finally, no significant effect of forage was found on plasma levels of polyunsaturated fatty acids or SPMs. On the other hand, the ratio of eicosapentaenoic acid to arachidonic acid was significantly higher in horses eating haylage compared with horses fed hay, steamed or not [42].

Dauvillier et al. [11] did not find a reduced risk of having mild asthma in horses eating soaked hay, nor a lower probability of having fungal elements in their airways. In that study, eating haylage did not reduce the risk of having mild asthma as opposed to the study by Olave et al. [42] The different results may be due to the diverse nature of both studies: feeding clinical trial [42] vs. prospective observational study [11], where all horses in the first study ate high-quality and controlled haylage, meanwhile in the second one, there was a large variation in the quality of products used.

Another option to decrease dust in hay is to apply treatments to it. A controlled trial in six asthmatic horses was performed to compare pelleted alfalfa hay (as this diet has been shown previously to normalise lung function in asthmatic horses) with oil-treated hay [66]. In the three months of the study, no differences in lung function, BALF cytology, or serum antioxidant enzymes were found between different feeds. All affected horses, in the oil and pellet groups, had a marked improvement in lung function and the oiled hay was palatable and well tolerated. The authors hypothesise that oil amalgamates fungal spores and other hay particles, and hence they are less frequently inhaled during hay feeding, similar to what happened in one study about liquid additives in concentrated food [52].

Different kinds of feeds could be an option to substitute dry hay. Each of them has advantages and inconveniencies that can vary depending on the owner’s perspective. For example, soaking hay is probably the cheapest method to decrease hay dust but it is very laborious, causes nutrient deficits, and is unstable in storage. Steaming, on the other hand, although easier, needs a high initial inversion and is not practical for a large group of horses. Finally, feeding haylage avoids the work involved in soaking or steaming hay and could be a good choice for horses. However, it has been reported in ponies that haylage induces a greater post-prandial insulinemic response than dry or soaked hay [67]. Therefore, haylage should probably not be fed to insulin-resistant horses. Future research should include, in our opinion, a thorough investigation regarding the effects of steamed hay on equine airways, as well as the pursuit of new possible treatments suitable for decreasing forage contamination and subsequent airway inflammation.

### 4.2. Effects of Stable Management and Bedding

The selection of bedding material is critical to enhance the quality of air within the stable and to avoid the risk of respiratory problems such as EA. Some authors have investigated the effects of different kinds of bedding on respiratory function. However, it is nearly impossible to isolate the effects of bedding from other factors such as feeding, as in many of the studies performed, both changes are applied together. 

Miskovic et al. [19] evaluated respiratory function and airway cytology in horses diagnosed with severe asthma and maintained in a low-dust environment (on pasture and not feeding on hay) for a long time. The results of this study showed similar clinical signs between asthmatic and non-asthmatic horses. No differences in BALF cytology or in standard lung function testing were found, but the absence of active inflammation is difficult to confirm without a measurement of inflammatory cytokines in BALF. However, some forced expiratory mechanics measurements (FEF90%, FEF95%, and FEF75-95%) were lower in asthmatic horses. The authors concluded that although the severely asthmatic horses maintained in low-dust environments were clinically indistinguishable from healthy horses, they continued to have evidence of peripheral airway obstruction, probably due to irreversible airway remodelling.

In relation to that fact, Leclere et al. [18] compared the effects of long-term (six months to one year) antigen avoidance (pasture and pelleted diet supplementation) and/or inhaled corticosteroids on chronic airway remodelling in horses with severe asthma. Lung function, airway inflammation, and smooth muscle remodelling in lung biopsies were assessed. The most interesting point of this high-quality study is that they focused on examining the effect of different treatments in airway remodelling, which is not common in clinical investigation. Their main conclusions are that airway smooth muscle, which increases in EA and limits lung function, significantly decreased by around 30% in horses treated with antigen avoidance. This finding confirms that chronic airway remodelling is partially reversible with only long-term antigen avoidance. However, treatment with inhaled corticosteroids can hasten this recovery. The results of these latter articles highlight the importance of a low-dust environment on airway inflammation and chronic airway remodelling. From a practical point of view, owners should be informed that although clinically sound, those horses are not completely recovered, and thus, they should be maintained in a controlled environment.

The effects of alterations in the stable environment on horses’ health have also been evaluated by measuring several analytes, indicative of airway inflammation in humans, in the breath of asthmatic horses. Ethane, carbon monoxide, and hydrogen peroxide were the analytes selected [68]. Changing from a bedding of straw and fed hay to a commercial dust-free bedding and dust-free forage caused not only a reduction in clinical signs of respiratory inflammation but also changes in breath ethane and carbon monoxide. On the contrary, hydrogen peroxide showed no significant differences under the two management systems.

Two recent studies compare the effect of peat as bedding, first with wood shavings [64] and then with three other bedding materials more often chosen nowadays: straw pellets, wood pellets, and another type of peat (loosely stored instead of baled storage) [61]. Peat is formed by a partial decomposition of vegetation or organic matter that happens in some natural areas. Its use is rather common in Nordic and Baltic countries where it is easily accessible and economical, but some environmental concerns about its extraction and utilisation exist, and its employment has been discouraged by some countries. Regarding the effect on horses exposed, only subtle differences were found among materials, as expected by the authors, because all horses in the study were healthy. No clinical signs appeared to be related to any of the materials, and only a higher mucus score was found during the straw pellet period [61,64]. Neutrophil percentages in both tracheal wash and bronchoalveolar lavage were also mildly increased in the straw pellet period compared with baled peat [61].

Another study also evaluates the differences between the effects of three bedding materials, straw, peat with shavings, and crushed wood pellets, on the equine respiratory function of healthy horses. All horses remained clinically healthy regardless of the type of bedding applied. There was not a clear influence of bedding on arterial blood gases or endoscopic evaluations of the respiratory tract, and only endoscopic scores of congestion were slightly worse in horses on straw [59]. Although all these articles show some relevant information concerning the effects of different kinds of bedding on respiratory function, their effect on asthmatic horses should be investigated before a firm recommendation about them can be made.

Increasing ventilation in stables is considered useful to diminish dust exposure. To investigate its effect on respiratory function, Walinder et al. [63] evaluated the consequences of mechanical ventilation in a horse stable on different parameters both in humans and horses. Their results showed that there was a decrease of nearly 50% in levels of CO_2_ with mechanical ventilation. Ammonia and ultrafine particles were also reduced. Airborne microorganisms and endotoxins, however, increased. The authors commented that it could be due to the minimal-velocity air flow that was operated due to climatic conditions, together with measurements taken during peak exposures (cleaning and feed delivery), which are less affected by low-to-moderate ventilation. Despite that, horses’ airway mucus and expression of interleukin-6 mRNA in BALF were significantly reduced after ventilation.

A clear conclusion that can be drawn from all these studies is that the objective of changes should not only be focused on forage or bedding but both aspects should be addressed at the same time. Additionally, using straw as bedding and dry forage for feeding should not be recommended for any stabled horse and it is especially contraindicated for asthmatic horses [11]. Nevertheless, there is an important lack of information concerning the influence of different kinds of bedding alone, without feeding changes, on equine airways, especially on asthmatic horses.

### 4.3. Use of Supplements

The administration of nutritional supplements as an adjuvant to EA treatment is an option commonly preferred by horse owners due to its feasibility. Some interesting research in this area is summarised below.

Supplementation with Omega-3 polyunsaturated fatty acids (PUFAs) results in an improvement in clinical signs (cough and respiratory effort), lung function, and BALF of both mildly and severely asthmatic horses [69]. The reason for this is that the incorporation of omega-3 PUFAs, at the expense of omega-6 PUFAs (precursors of arachidonic acid), decreases the substrate available to synthesise proinflammatory eicosanoids [70]. Omega-3 PUFAs have also been implicated in the modulation of inflammation by decreasing inflammatory cytokines such as tumour necrosis factor-α, interleukin-1ß (IL-1 ß), IL-6, and IL-8 [71]. Evidence shows that docosahexaenoic acid in plasma (DHA), as a consequence of fatty acid supplementation, is responsible for clinical improvement [69]. Thus, supplementation with Omega-3 PUFAs could be an additional tool to best manage EA.

Another supplementation alternative involves the addition of sunflower oil or seal blubber oil to asthmatic horses’ diets [72]. Those products are well accepted by horses and their amount of plasma fatty acids varies in response to dietary fatty acid content. Also, the daily intake of long-chain Omega-3 PUFAs was incorporated into the plasma and leukocyte membrane phospholipids of the studied horses. Likewise, a decrease in pulmonary epithelial lining fluid (PELF) was shown, and after the intake of seal blubber oil, the count of total PELF cells was similar to or below the range reported in healthy horses. Due to that, the authors suggested a correlation between the consumption of seal blubber oil and an improvement in the asthma condition. Nevertheless, more studies are required before the clinical efficacy of this diet supplementation can be assessed properly.

Other nutraceutical and herbal supplements for horses with EA have been studied, among other reasons, to avoid using pharmacological products forbidden in races or other sports. One nutraceutical product, a complementary feed containing maltodextrin, calcium carbonate, *Arthrospira platensis* (12%), and fermented pineapple (5%), supplemented with antioxidants, was studied recently [73], and it seems to improve the clinical score and mucus production. However, there were clinical differences between the nutraceutical and control groups at the beginning of the study that may have influenced these results. Other previous studies showed no significant effects on the clinical signs or respiratory cytology of horses given herbal products [74,75].

Insufficient evidence-based findings exist, in our opinion, to support the usage of nutraceutical and herbal supplements in horses with asthma apart from supplementation with Omega-3 PUFA. However, more research in this area would be interesting because finding a useful product could decrease the need for medication for this chronic disease. 

## 5. Owners’ Adherence to Treatment Recommendations

Significant emphasis has been placed on identifying the most effective environmental modifications for asthmatic horses. However, there are limited data on the practicality and feasibility of implementing these recommendations among horse owners. Two interesting and recent studies address this relevant topic. In the first study, Boivin et al. [76] investigated which of the recommendations for horses with mild asthma given to the owners were implemented. Recommended measures included changing feeding practices, decreasing exposure to dust by ventilation, and increasing the hours that horses spend outside or changing bedding. Additionally, owners were also recommended to administer inhaled or systemic corticosteroids and bronchodilators if necessary. A wide range of months after diagnosis (2–35), information was collected by telephone surveys regarding adherence to those recommendations.

All owners who answered the telephone survey reported that they implemented at least one of the measures recommended. Most of them avoided feeding dry hay and changed to soaked hay (>50%), pellet hay, or others. Also, nearly 80% of the owners increased the hours that the horses were turned out. However, diet and turnout management changes were only in place temporarily and only 30% of the respondents described rigorous and permanent changes in management.

Concerning the response to measures and treatment, more than 70% of the owners described a positive effect. Nevertheless, more than 50% of them received oral or inhaled corticosteroids. There was no relation between any of the measures and the outcome of the horses but an association with clinical signs was not found either.

Significant limitations of that study included a lack of standardisation in treatment combinations, that the evaluation of outcome was performed by owners (risk of bias because of selective memory), and the absence of enough horses with poor outcomes that caused a lack of power to evaluate the results. Another limitation of that study is linked to the period in which those recommendations were given, as steamed hay as well as Omega-3 supplementation were not common in those years.

Another recent study in this regard, performed by Simões et al. [22] had a similar objective although used some different methods. In the first place, the authors fixed a moment to evaluate the effect of recommendations. It was established one year after a diagnosis of severe asthma was given to the horse and after an initial questionnaire about environmental management was obtained through the owner. The initial questionnaire also evaluated similar aspects to the work by Boivin et al. [76] (“housing, feeding conditions, triggers and clinical signs”), and then six management recommendations were provided (“ventilation, type of housing, time outdoors, removal of the horse while cleaning, bedding and forage”). After one year, a similar telephone interview was performed to compare with previous results. In this report, compliance of the owners was not considered as good as in the previous study but many of the horses showed an improvement, and no medication was still needed in almost half of them. Comparisons, however, should be performed cautiously because, in a study by Simões et al. [22] poor compliance was considered when only two or fewer recommended measures were taken (more than 50% of the owners in this study), although in a study by Boivin et al. [76] a positive response was reflected when at least one of the measures recommended was implemented, which is probably a very limited objective.

As a result of both studies, we can conclude that in most cases, owners implemented some of the measures recommended, in general, those less demanding for them, such as changing their diet or increasing turnout time, and in many cases, only for a short period of time. Soaking hay and keeping horses at pasture were the least popular measures in the study conducted by Simões et al. [22], however, in a study by Boivin et al. [76], owners preferred soaking the hay and keeping horses outside for more hours to other kind of measures. Despite that, many of the horses improved and needed less medication. It can be inferred from this that if we can effectively communicate the importance of these modifications to owners and improve their adherence, a huge improvement could be expected in most horses, together with an important decrease in medication.

## 6. Conclusions

The environmental management of equine asthma is a considerably complex problem to address. However, our responsibility as clinicians is to help horse owners understand the benefits of current recommendations, together with the best approach to implement them. Feeding alternatives to traditional dry hay, which is not deemed suitable for asthmatic horses, include soaked or steamed hay or haylage. All of them achieve a decrease in dust quantity and airborne contamination, provided they are well prepared and conserved. Nevertheless, the improvement in hay quality obtained does not always correlate with a benefit in equine airways, especially in the case of steaming, and this should be further investigated. Soaking hay, and to a lesser extent steaming it, alters the nutritional value of hay and its palatability, causing particular mineral and protein losses. Consequently, diet supplementation is needed, and further studies regarding dietary modifications for these horses are necessary. With regard to bedding, an alternative to straw is crucial. Although numerous options have been investigated, no consistent differences among them have been reported. The use of different study designs, evaluating only the effects of bedding, will help to determine the best option to elect. Finally, the management of EA in horses is demanding and frustrating. Therefore, most of the recommended measures are not implemented by owners in the end, or in many cases, they are only temporarily in place. For this reason, the focus of EA management should be a detailed explanation to owners of the real advantages of these measures to ensure that they are applied.

## Figures and Tables

**Table 1 animals-14-00446-t001:** Comparison of types of forages regarding their quality analysis. DM: dry matter.

	Type of Forage			
	Dry Hay	Soaked Hay	Steamed Hay	Haylage
Dust exposure	High respirable dust exposure [38,42]	Decrease in the number of small particles [43,44]	Significantly reduces particles [40,44,46]	Respirable dust exposure significantly lower than in hay [38,40,42]
Fungi presence	Increases probability of fungi in tracheal wash of horses with mild asthma [11]	Mould counts decrease after soaking [40,43]	Decreases fungi content [11]. Nearly no typical fungi detected [40]	Lower mould counts compared with hay [11,43]
Endotoxin	High endo-toxin expo-sure [38,42]	Results affecting only this change not described in the consulted bibliography	Conflicting results: No differences between steamed hay and haylage [47]. Higher than in haylage [42,48]. Lower levels than hay [48]	Conflicting results: Lower endotoxin exposure than hay and steamed hay [42]; no differences among forages in another study [47]
Microbiological counts	High count of bacteria [40,41]	Lower levels of bacteria than in hay [40]. Increase in total bacterial counts [44,45]	Nearly eliminates microbial contamination [40,44,46]	Excessive microbial counts more prevalent in haylage samples, especially when DM content is not adequate [41]
ß-glucans	Discrepancies in levels among studies [25,42,47]	Results affecting only this change not described in the consulted bibliography	Conflicting results: No differences with hay [42,48] or haylage [42]. Higher levels than haylage [47]	Conflicting results: No differences among forages [42]. Fewer ß-glucans in the haylage compared with straw [47]

## Data Availability

No new data were created in this study. Data sharing is not applicable to this article.

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
