# Peer review of "Environmental Management of Equine Asthma"

_animals, 2024, doi:10.3390/ani14030446_

Round 1

Reviewer 1 Report

Comments and Suggestions for Authors

This review paper concerns itself with environmental factors associated with dust in the air, one of the primary causes of equine asthma (EA),  and techniques to lessen this.

It is structured as a narrative review- this feels a bit simplistic, and I would recommend to the authors to involve a bit more critique in their text, as it is clear they have studied the available literature well and are equipped to do this. The information needs to flow towards a conclusion, beyond a simple description and to include a degree of analysis- I would appreciate critic based on the lit review, as well as identification of key problems with suggestions of future research for each section, for now there are only some tentative recommendations that are quite weak. Also, these recommendations should be highlighted, as they can get lost in the text. If a person is looking for a review, they want to also find the relevant information easily.  The conclusions and recommendations at the end are slightly general and need to outline clearly where need for future research lies. 

The review is structured well and follows a logical flow on how the information is presented. 

line 126, "they" who? Which of the two types of particles, it is not clear

line 132 incorrect name, it is Thermoactinomyces (not Thermacintomyces)

lines 210-214 and 397-398 describe steaming and soaking, please detail what these entail. One sentence suffices but not every reader is familiarized with these procedures. Not only that, but they are not standardized and each study has followed different protocols. 

line 166- what does "our control" mean? please rephrase

lines 413-415 what ARE the effects of steaming over dust? A specialized reader will be familiar but if you are reviewing the lit on steaming, this is information that should be taken into account in the recommendations that you are making. 

line 459- regarding insulin resistance and obesity- you have said nothing so far about the two, and this info comes to me suddenly- maybe explain in a short sentence why haylage is not recommended for insulin-disregulated animals (pronounced insulin response, glycemic response, citation needed, for example Carslake 2018)

line 487 equine breath of horses is quite redundant 

line 513- what do you mean endoscopy results were worse? do you mean the mucus scores?

line 549- which nutraceutical product?

Comments on the Quality of English Language

The quality of English is quite problematic, several mistakes might well be due to haste, but the entire paper needs to be proofread with a lot of attention, some examples are below. It is a bit distracting while reading and some sentences are difficult to understand. 

line 39- denomination usually refers to money or religions. Do you mean definition or maybe nomenclature?

line 54-55 this sentence is very difficult to read, it is forced and seems to miss something, or not be linked with something. Please rephrase

line 64 do you mean "the" latter, not "this"?

line 76-77 please rephrase

line 90 should read not related "to" veterinary medicine

line 128 - "these" not this

line 131 "succeed" not success

line 140 diagnosed with, not of

line 168 compared to, not with

178 hay is the most common, not commonest

line 183 which requires time

line 259- attain means "to reach", do you mean pertain to/ refers to?

etc, etc. There are multiple instances of this, please revise. 

Reviewer 2 Report

Comments and Suggestions for Authors

The review article on Environmental management of Equine Asthma appears to be a balanced review on a wide selection of published studies.

I suggest to the authors to formulate a table of summary findings & recommendation/ conclusion for each of the factors reviewed, for example, forage quality analysis, effects of hay treatments over its nutritional quality and acceptance, etc.

This summary table can help readers to make easy reference and understanding on the pros and cons for each factor examined.

Reviewer 3 Report

Comments and Suggestions for Authors

I found the article very interesting, but suggest edits for clarity and readability.

Please double-check all statements and properly cite sources throughout the manuscript.

Line 37-42: “Equine Asthma (EA) is a chronic non-infectious inflammatory disease of lower air-37 ways that is characterized by respiratory signs such as cough and increased respiratory 38 effort. Denomination of this disease has recently changed. Two previously considered as 39 different clinical entities: recurrent airway obstruction (RAO) and inflammatory airway 40 disease (IAD) are now denominated as EA (severe or mild respectively). They have simi-41 lar clinical presentations, but many differences may exist in causes, severity, and patho-42 logic characteristics.” Please cite.

Line 46-47: I suggest, adding how these factors affect asthma, would make article more interesting.

Line 176:177: Evidence suggests that feeding is the most important environmental factor causing 176 airway inflammation in horses.

531-533: “Supplementation with Omega-3 polyunsaturated fatty acid (PUFA) results in an improvement in clinical signs (cough and respiratory effort), lung function and BALF of both mild and severely asthmatic horses”. Please add the underlying mechanism.

Please make the conclusion short and to the point.

Round 2

Reviewer 1 Report

Comments and Suggestions for Authors

Dear authors, thank you so much for investing the time and effort into these improvements and changes, I truly believe the manuscript is now an enhancement to the current knowledge. I particularly enjoyed adding the knowledge gaps for each chapter and the summarizing tables for different findings at the suggestion of another reviewer. It is a massive improvement and makes for a good read and comprehension. I recommend publishing of this manuscript with no other changes.